# Systematic Review and Meta-Analysis of Dietary Interventions and Microbiome in Phenylketonuria

**DOI:** 10.3390/ijms242417428

**Published:** 2023-12-13

**Authors:** Francesca Ubaldi, Claudia Frangella, Veronica Volpini, Paola Fortugno, Federica Valeriani, Vincenzo Romano Spica

**Affiliations:** 1Department of Movement, Human, and Health Sciences, University of Rome “Foro Italico”, 00135 Rome, Italy; f.ubaldi@studenti.uniroma4.it (F.U.); frangellaclaudia@gmail.com (C.F.); v.volpini@studenti.uniroma4.it (V.V.); vincenzo.romanospica@uniroma4.it (V.R.S.); 2Department of Human Sciences for the Promotion of Quality of Life, University San Raffaele, Via di Val Cannuta 247, 00166 Rome, Italy; paola.fortugno@uniroma5.it; 3Human Functional Genomics Laboratory, IRCCS San Raffaele Roma, Via di Val Cannuta 247, 00166 Rome, Italy

**Keywords:** inborn errors of metabolism, phenylketonuria, dietary interventions, microbiota, dysbiosis

## Abstract

Inborn errors of metabolism (IEMs) comprise a diverse group of monogenic disorders caused by enzyme deficiencies that result either in a toxic accumulation of metabolic intermediates or a shortage of essential end-products. Certain IEMs, like phenylketonuria (PKU), necessitate stringent dietary intervention that could lead to microbiome dysbiosis, thereby exacerbating the clinical phenotype. The objective of this systematic review was to examine the impact of PKU therapies on the intestinal microbiota. This research was conducted following the PRISMA Statement, with data from PubMed, Scopus, ScienceDirect, and Web of Science. A total of 18 articles meeting the inclusion criteria were published from 2011 to 2022. Significant reductions in several taxonomic groups in individuals with PKU when compared to the control group were detected in a quantitative analysis conducted across seven studies. The meta-analysis synthesis indicates a contrast in biodiversity between PKU subjects and the control population. Additionally, the meta-regression results, derived from the Bacillota/Bacteroidota ratio data, suggest a potential influence of diet in adult PKU populations (*p* = 0.004). It is worth noting that the limited number of studies calls for further research and analysis in this area. Our findings indicate the necessity of enhancing understanding of microbiota variability in reaction to treatments among PKU subjects to design tailored therapeutic and nutritional interventions to prevent complications resulting from microbiota disruption.

## 1. Introduction

Inborn errors of metabolism (IEMs) are heritable monogenic disorders that often show multi-systemic involvement since the neonatal period. Typically, gene mutations in IEMs lead to inadequate enzymatic or transport activity. This condition leads to the buildup of harmful substances or a lack of necessary nutrients [1]. IEMs may be concerned with protein, carbohydrate, or fat metabolism, or they may be related to altered organelle function, manifesting as complicated medical conditions involving multiple organs and systems [2]. The treatment of these conditions depends on the metabolic defect: in some conditions, for example, phenylketonuria (PKU), specific dietary treatment aimed at reducing the buildup of toxic metabolites in the body is available, and it is the main and crucial intervention. However, the metabolic dysfunction induced by the IEM, as well as the dietary strategies used to alleviate it, affect the gut microbiota, resulting in exacerbation of the clinical phenotype [3]. 

PKU is a rare autosomal recessive disorder mainly resulting from mutations in the gene that encodes the enzyme phenylalanine hydroxylase (*PAH*). A few cases of PKU (1–2% of all patients with hyperphenylalaninemia) are caused by defects in BH4 metabolism or pathogenic variants in the DNAJC12 gene [4]. Phenylalanine hydroxylase is a tetrameric, iron-containing monooxygenase that converts L-phenylalanine (Phe) into L-tyrosine (Tyr) through hydroxylation. This process primarily takes place in the liver but also happens in the proximal renal tubules, and it requires molecular oxygen, iron, and tetrahydrobiopterin (BH4) as a cofactor. BH4 oxidizes to quinonoid dihydrobiopterin (qBH2) during the hydroxylation reaction and then is enzymatically transformed back to its BH4 form so that it can continue supporting the phenylalanine hydroxylation. 

Since about 90% of the daily intake of phenylalanine must be metabolized through this route, defects in phenylalanine hydroxylase or other components of this pathway cause the accumulation of phenylalanine in the body. The condition of hyperphenylalaninaemia (HPA) is easily detectable in the blood. The blood tyrosine concentration is also found to be decreased but hypothyrosinemia is less serious, possibly due to dietary Tyr intake. The deamination of phenylalanine in an HPA condition results in the production of phenyl ketones, including phenyl pyruvate, which are excreted in the urine, from which the name “phenylketonuria” originates [5]. 

The severity of the disease depends on the level of residual enzymatic activity. If there is complete or near-complete loss of phenylalanine hydroxylase activity, it is classified as classic PKU, whereas less severe forms are classified as moderate PKU, mild PKU, mild HPA, or benign HPA [6]. The main treatment strategy for PKU is a low phenylalanine diet combined with phenylalanine-free L-amino acid supplementation. Dietary treatment is very effective because the body cannot synthesize phenylalanine, which is an essential amino acid, and consequently, its blood concentration is closely related to the amount of phenylalanine introduced in the diet. Some patients with BH4 responsiveness can be treated with sapropterin dihydrochloride (Kuvan, BioMarin Corporation, Tiburon, CA, USA), which is a pharmaceutical chaperone. Moreover, the FDA approved a new therapy in 2018 for adults in the USA and for patients aged 16 years or older in Europe. This therapy is based on a bacterial enzyme that can break down Phe (PEGVALIASE PQPZ, BioMarin Pharmaceutical Inc., Novato, CA, USA) [4]. Additionally, enzyme replacement and gene therapy are potential future treatments [7,8].

Diet therapy is still the main treatment for PKU management and can include three aspects: (i) restriction of natural protein intake, (ii) consumption of low-protein food products, and (iii) supplementation with phenylalanine-free amino acids [9]. The main objective is to avoid any neurological complications by maintaining blood phenylalanine levels within the therapy target range, while also meeting nutritional requirements to ensure regular growth and maintain the body structure [5,8]. Phenylalanine is essential for protein synthesis, and, therefore, a minimal amount must be provided through the diet to support tissue growth and repair during childhood and tissue repair in adulthood, maintaining plasma phenylalanine concentrations within recommended ranges (2–6 mg/dL or 120–360 μmol/L) [8,9,10,11]. To replace natural protein, a solution containing a mix of phenylalanine-free amino acids enriched with tyrosine can be used as a substitute [9,10]. Protein blends with low or no phenylalanine content are successfully used as an alternative dietary supplement [5]. 

Glycomacropeptide (GMP) is a component of whey protein, which does not contain Phe. Studies suggest that its consumption can increase feelings of fullness and reduce inflammation. However, GMP, in addition to Phe, is also low also in some other large neutral amino acids (LNAAs), such as tryptophan, leucine, methionine, and histidine, that play an important role in blocking Phe transport across the intestinal tract and the blood-brain barrier (BBB) by competing for the amino acid transporter LAT1. On the other hand, elevated Phe levels hinder other LNAAs from crossing the BBB, causing their deficiency in the brain. Supplementation with a mixture of LNAAs should, therefore, be considered as an alternative dietary supplement in combination with GMP or some other low Phe-free diet [6,9,10,11,12]. The goal of dietary treatment for IEMs, and specifically for PKU, is to decrease the accumulation of toxic substances in the body. However, this treatment can also lead to excessive loads of some nutrients, nutrient deficiencies, or both, which can be dangerous and linked to the risk of developing non-communicable diseases [11,12,13]. 

Advancements in research are providing new treatments for PKU patients, like engineered probiotics that express enzymes to metabolize phenylalanine [14]. However, there is a need for increased knowledge of the interaction between the intestinal microbiota and nutritional therapy [14]. The available data suggest that the microbiome can be positively or negatively influenced by nutritional, pharmacological, or combined interventions for patients with PKU, as well as what occurs for other IEMs [15,16,17,18,19,20]. Therefore, it is crucial to investigate the metabolic role of the gut microbiota in PKU patients [16,17]. This could lead to improvements in the intervention strategies by promoting beneficial pathways and inhibiting harmful ones, and allowing for personalized biotherapeutics.

The aim of this systematic review was to examine the microbiota structure in PKU patients by analysing existing literature on the topic. A meta-analysis was also conducted to determine changes in biodiversity indices in the gut microbiota of PKU patients compared to the control group. Additionally, specific modulators of the microbiota biodiversity and composition were investigated whenever possible.

## 2. Materials and Methods

### 2.1. Selection Protocol and Search Strategy

The present systematic review and meta-analysis was performed based on the PRISMA guidelines [21], and it was registered in PROSPERO [reference number CRD42020177972]. 

The PICOS framework was used to formulate the review question and the eligibility criteria: (a) Population: PKU patients; (b) Intervention: diet and/or pharmacological treatment; (c) Comparison: PKU to a control group; (d) Outcomes: the abundance of phyla, number of OTUs (operation taxonomic units), Bacillota/Bacteroidota ratio, and alpha and beta diversity indices; (e) Study: descriptive, retrospective, and prospective studies. Four electronic databases (PubMed, Scopus, ScienceDirect and Web of Science) were interrogated using the following terms: (“metabolism, inborn errors”[MeSH Terms] OR (“metabolism”[All Fields] AND “inborn”[All Fields] AND “errors”[All Fields]) OR “inborn errors metabolism”[All Fields] OR (“inborn”[All Fields] AND “errors”[All Fields] AND “metabolism”[All Fields])) AND ((“phenylketonurias”[MeSH Terms] OR “phenylketonurias”[All Fields] OR “phenylketonuria”[All Fields]) AND (“microbiota”[MeSH Terms] OR “microbiota”[All Fields] OR “microbiotas”[All Fields] OR “microbiota s”[ All Fields] OR “microbiotae”[All Fields] OR “microbiome”). The databases of PubMed, Scopus, and Web of Science were searched using the title, abstract, and MeSH terms for the former and topic by title, abstract, and keywords for the latter two. The search encompassed all relevant literature published from the start of the databases until 30 September 2022. 

### 2.2. Inclusion and Exclusion Criteria 

From the databases, the references were imported into Covidence, a platform for managing systematic reviews, for relevance assessment. Next, titles and abstracts were screened according to the inclusion criteria. Four authors performed the screening and read the full text independently (C.F., F.V., F.U., and V.V.). Disagreements were resolved by consensus. Systematic reviews, randomized controlled trials, cohort studies, case–control studies, cross-sectional studies, and narrative reviews were included, as well as studies in which the gut microbiota had been analysed from faecal samples by cultural methods, 16S rRNA genome sequencing (16S amplicon sequencing), or by PCR-based molecular methods. Reviews and meta-analyses were assessed to identify further articles in their references. Only articles published in the English language were included.

### 2.3. Data Extraction Process and Quality Assessment 

The data extracted from each eligible record included bibliographical information, study design, patient/population characteristics, study inclusion/exclusion criteria, treatment/dietary control/diet type, and microbial profile analysis. 

The quality assessment was performed using the NOS (Newcastle–Ottawa Quality Assessment Scale). Three researchers (C.F., F.U., and V.V.) employed the NOS to evaluate the quality of the literature, with a fourth reviewer intervening to resolve discrepancies as required (F.V.). The NOS comprised three criteria: selectivity, comparability, and outcome [22]. A study with a score of ≥7 was considered good quality, 5–6 fair quality, and 0–4 low quality.

### 2.4. Data Synthesis 

To examine variations in the gut microbiota composition, we performed a literature review that provided a descriptive analysis while accounting for discrepancies in microbiota evaluation, the small number of participants, and the limited data and quality of the collected research. We used Comprehensive MetaAnalysis (https://meta-analysis.com/, 6 December 2023) to conduct meta-analyses. We calculated standardized mean differences (SMDs) between PKU patients and controls for bacterial diversity indices such as the number of OTUs (operation taxonomic units), the Bacillota/Bacteroidota ratio, the alpha diversity, and the Shannon index. In cases where there were errors regarding the unit of analysis, we only divided by the total number of participants in the control group and left the means and standard deviations unchanged (http://www.handbook.cochrane.org, 6 December 2023). In one article, only a cohort of adults was selected, excluding data from non-compliant subjects to reduce heterogeneity [23]. The 95% confidence interval (95% CI) was also calculated. A random-effects model was utilized to form a combined computation of the SMD, and a fixed-effects model was employed to ascertain robustness and potential outliers. To evaluate statistically significant heterogeneity, we used the I^2^ (the variation reflecting the true heterogeneity in a percentage), τ^2^ (the random-effects between-study variance), and the *p*-value from Cochran’s Q test. *p* < 0.05 was considered statistically significant. Funnel plots were also utilized to explore potential publication bias. Subgroup analyses were conducted for primary and secondary outcomes that were reported in two or more studies within each subgroup [22,24,25]. Meta-regression and subgroup analyses were utilized to investigate the expected sources of heterogeneity [26,27]. Predefined subgroup analyses were conducted, including intervention types such as supplements or dietary interventions, sample size, ages, publication year, methodological quality of the study, and WHO region of origin. To execute a sensitivity analysis, individual studies were excluded or the effects model was adjusted.

## 3. Results and Discussion

While the dietary treatment for IEM aims to lower toxic compounds, it can also lead to hazardous nutrient deficiencies [28]. Therefore, it is essential to take into consideration the microbiota structure in patients with PKU and its alteration following specific therapies. Here, the purpose of the present systematic review was to explore the structure of the microbiota in patients affected by PKU, considering the specific modulators of microbiota biodiversity and composition, by analysing the data coming from the available literature on this topic.

The flow diagram of the study selection process for this systematic review is described in Figure 1.

A total of 777 studies were found across all of the searched databases. Following duplicate removal, 294 articles were available for further analysis. Among them, 254 studies were ruled out upon reviewing their titles and abstracts. Based on the inclusion and exclusion criteria, 40 full texts were evaluated and 22 of them were excluded for not meeting the inclusion criteria. Finally, the qualitative analysis included 18 studies, comprising 12 original studies (Table 1) and 6 reviews (Appendix A).

The included articles were published between 2011 and 2022, and the studies were performed in several countries. Three studies were conducted in Italy [23,30,39], three in the USA [32,33,35], two in the Netherlands [37,38], one in the UK [31], one in Brazil [34], one in China [36], and one in Jordan [29].

In the qualitative analysis, we also included six reviews (Appendix A). Two narrative reviews were exclusively focused on PKU [19,40], while the other papers summarized the relationship between the microbial profile and IEM patients in general. None of these reviews were performed following systematic approaches [3,16,17,18,19,40].

Based on the results obtained from the already available narrative reviews, a meta-analysis was performed, focusing on seven studies. They were selected from among the 12 that could be included in a systematic review (Table 2). Only studies of humans with PKU using 16S amplicon sequencing techniques were included in the meta-analysis. Two studies on mouse models [35,38] and three studies that used techniques other than 16S amplicon sequencing were eliminated (Figure 1) [29,31,39].

The included articles were published between 2016 and 2022 and performed in several countries. Two studies were conducted in Italy [23,30], two in the USA [32,33], one in Brazil [34], one in China [36], and one in the Netherlands [37].

In all of these studies, the analysis was carried out on faecal samples. Illumina technology was used in six of the seven papers [23,30,32,35,36,37], and only Pinheiro de Oliveira et al. (2016) used the Ion Torrent platform [34]. Four studies used Illumina MiSeq [23,30,32,37], one Micro MiSeq [33], and Su et al. (2021) used a combination of data from Illumina MiSeq PE300 and NOVAseq PE250 [36]. Almost all of the studies focused on the V3–V4 region of the 16S amplicon, while Mancilla, Pinheiro De Oliveira and MacWorther used custom-designed primers that amplified both Archaea and Bacteria [32,33,34].

In addition to the analysis of the methodology used, data were also extracted for the study of biodiversity (Table 3). To comprehend the data’s significance, it is essential to begin with defining the biodiversity indicators applied. The scientific community universally employs indices from ecology obtained in human and environmental microbiota analyses, known as alpha and beta diversity. Several measures describe the microbiota in a sample. These measurements do not provide information regarding changes in specific taxa abundance but instead enable speculation on more expansive changes or variations in the microbial composition of the population. The various measurements reflect either the richness (quantitative measures like Shannon’s index or the number of OTUs) or distribution (uniformity measures such as evenness, equitability, or Simpson’s index) of the microbial sample or a combination of both (Chao1). Alpha diversity measures microbiota diversity within a single sample, while beta diversity measures the similarity or dissimilarity between two communities. There are various indices for measuring beta diversity, each reflecting different aspects of community heterogeneity. Key differences relate to how the indices assess variations in rare species, whether they consider only the presence/absence of microorganisms or incorporate information about their abundance, and how they interpret shared absence. Bray–Curtis dissimilarity, on the other hand, is the most widely used measure that considers both the size (overall abundance per sample) and shape (abundance of each taxon) of communities [41]. 

In all seven papers included in the meta-analysis, the Shannon index was used to assess biodiversity, and in five of the seven papers, the index decreased in subjects with PKU [30,33]. In McWhorter et al., 2022, the Shannon, Chao1, and Simpson parameters seemed to increase rather than decrease in PKU patients [33]. However, we must consider that these are subjects with extensive age variability. In addition to being affected by changes in diet and frequent antibiotic use, the microbiota is also profoundly affected by age [41]. Several factors can cause changes in the bacterial makeup of the host’s gastrointestinal tract over time [42,43,44]. Levels of microbiota-derived metabolites are higher in older individuals with age-related illnesses and cognitive impairments compared to younger, healthy age groups [45,46,47,48]. Therefore, a comprehensive comprehension of alpha diversity necessitates more than simply measuring it, and contextual information is required for purposeful evaluations. Further analysis using age-stratified data (Figure 2A) confirmed this trend. The Shannon index tended to decrease in subjects with PKU in both age subgroups, but the change was not significant (for childhood SMD  =  −1.669; 95% CI −11.619 to −8.282, *p*  = 0.060; and adult SMD  =  −1.097; 95% CI −3.548 to −1.353, *p* = 0.138). From the point of view of indicators of beta diversity, in all of the studies, a difference was shown between the microbial compositions of the subjects compared to the control, while in two papers, MacWorker et al. and Montanari et al., the microbial communities were similar (Table 3). It is likely that the variation in the age of the cohort is a strong confounding factor, as is the treatment in MacWorker et al., which allowed for a higher protein tolerance and normal dietary behaviour [33].

The Bacillota/Bacteroidota ratio (formerly Firmicutes/Bacteroidetes), that is often used to describe gut health was also analysed. In fact, it has been proposed as a possible biomarker of dysbiosis since it has been reported by several studies that there is a difference in this index between normal weight versus obese individuals [47]. Consequently, the Bacillota/Bacteroidota ratio is often cited in the scientific literature as a hallmark of obesity or less than ideal health states, although the scientific community is not unanimous on this definition. In the present study, as shown in Table 3 and Figure 2B, the Bacillota/Bacteroidota ratio tends to decrease in subjects with PKU; however, the change is not significant except in childhood (for childhood SMD  =  −1.511; 95% CI  −2.524 to 0.0–499; *p* = 0.029 and adult SMD  =  −1.159; 95% CI  −2.368 to 0.049, *p* = 0.060).

Figure 2C also shows the pooled estimate for OTUs (for childhood SMD  =  −10.583; 95% CI  −13.792 to −7.374, *p* > 0.001 and adult SMD  = −6.394; 95% CI  −10.947 to −1.841, *p* = 0.035), with a significant difference between groups.

Table 3 illustrates the patterns of the primary phyla and genera in the seven studies. Technical terms and abbreviations have been defined in their first use, and the language used throughout the text is unbiased, objective, and grammatically correct. Distinguishing features are apparent in the two most prevalent phyla, Firmicutes and Bacteroidetes.

In their 2019 study, Bassanini et al. [30] compared gut microbial communities in children with PKU and mild hyperphenylalaninemia (MHP) following an unrestricted diet. That study found that the relative abundance of Firmicutes and Bacteroidetes was similar in both groups, with a slight decrease observed in the PKU group. Furthermore, PKU subjects exhibited a reduction in *Bacteroides* and *Prevotella*, which are the primary genera of the Bacteroidetes [48]. This change can be attributed to the dietary treatment type, which is defined by a greater intake of carbohydrates and fibre alongside a significantly lower consumption of protein. Consequently, the gut microbiota of PKU subjects has diminished in butyrate-producing species and increased in genera like *Blautia*, which have a pro-inflammatory impact on the intestinal mucosa. Studies have shown that *Blautia* spp. can stimulate cytokine production, including tumor necrosis factor-alpha (TNF-alpha), which is linked to the acute phase of the immune response [49].

In Pinheiro De Oliveira et al. 2016, the gut microbiota of PKU patients with Phe-restricted dietary treatment was compared with that of healthy subjects. In both groups, the predominant phyla identified were Bacteroidetes and Firmicutes. PKU patients exhibited an increase in *Prevotella*, *Akkermansia*, and Peptostreptococcicee. Notably, *Akkermansia muciniphila* was identified at low concentrations in obese individuals and was inversely correlated with body weight in both rodents and humans [50,51]. This mucin-degrading bacterium has been linked with health benefits and a decrease in cardiometabolic disorders [52]. The increase in this bacterium in those affected is likely related to a diet predominantly composed of carbohydrates (about 80%), and because their consumption of protein came from protein substitutes enriched in calcium, phosphorus, iron, manganese, and zinc, and devoid of Phe, while their selenium intake remained low [34].

In McWhorter et al. 2022, the data showed that patients receiving pegvaliase-pqpz consumed a lower percentage of kilocalories from total protein and lower amounts of micronutrients, but they consumed higher amounts of protein. Microbiome data reported in this study revealed higher abundance of the phylum Verrucomicrobia and genus Lachnobacterium in the group following traditional dietary restriction and a higher abundance of the genus *Prevotella* in the Pegvaliase pqpz-treated group. Analysis of the microbiota of the group not taking the drug showed a greater presence of microorganisms involved in the pathways of carotenoid and amino acid metabolism. In addition, an increase of *A. muciniphila* in PKU subjects under dietary treatment also emerged [33]. In Montanari et al., 2022, the effect of 6 months of dietary supplementation with GMP on the bacterial composition was evaluated. No substantial changes were observed; however, a specific prebiotic effect of the genus *Agathobacter* spp. and, less so, *Subdoligranulum*, was detected. An examination of bacterial groups displayed a diminutive decline in the Verrucomicrobia phylum after 6 months, although it was not statistically significant. Additionally, there was a tendency towards a rise in Firmicutes while consuming GMP. In comparison, after GMP dietary treatment, there appeared to be a reduction in *Akkermansia*, but it was not statistically significant, and the *Escherichia-Shigella* count was greater [33].

Su et al. (2021) demonstrated the reduced presence of the Bacteroides genus in the PKU group, which was also connected with negative blood phenylalanine (Phe) levels [36]. However, there was considerable variation in blood Phe concentrations, suggesting inadequate dietary control among the monitored individuals in this study. This aspect raises doubts about the obtained data, underlining the fundamental importance of adequate patient selection to plan studies in this population.

Timmer et al., 2021, found an increase in several members of Lachnospiraceae in the faecal microbiome of adult PKU patients (19–50 years old) and a decrease in Ruminococcaceae [37]. An enhanced abundance of Lachnospiraceae was also observed in the faecal microbiome of pigs fed a diet comparable to that used with PKU patients: a natural low-protein diet supplemented with amino acids [46]. However, the data regarding Lachnospiraceae differs from the results of Pinheiro de Oliveira [34], in which all the patients were children, and some of them had used antibiotics in the 6 months prior to faecal sample collection. Antibiotics could explain the conflicting results. It should also be pointed out that the microbiome of the children was different from the adults [53,54].

To explore the impact of moderator variables on the microbiota biodiversity in PKU subjects, subgroup analyses and meta-regressions were performed. Data were collected on the Bacillota/Bacteroidota ratio, the OTU, the alpha diversity indices, and Shannon, and we carried out a meta-regression of the included intervention types (supplements or dietary interventions), sample size, the publication year, the methodological quality of the study, and the WHO regions origin. Only in the case of the Bacillota/Bacteroidota ratio did the meta-regression show that the type of treatment influenced this ratio in adults (*p* = 0.004), and this was also observed, although not significantly, for drugs (*p* = 0.410). 

The implementation of neonatal screening and early intervention has eliminated severe and permanent cognitive impairment among individuals with PKU [55]. Adults diagnosed with the condition require further investigation. Over the past four decades, the literature has revealed that discontinuing treatment during the developmental stages and during adulthood has adverse effects [56,57]. It is crucial to determine whether patients should continue their treatment during adulthood and which type of treatment is the most effective. Currently, no research distinguishes the effects of Phe levels or treatment on microbiota during different stages of life, such as childhood, adolescence, or adulthood. According to the subgroup analysis (Q = 2.10, df = 1, *p* < 0.05), the study’s location has a minor moderating effect. The literature suggests that the human microbiota is significantly influenced by diet [16,17,58]. Recent scientific and medical advancements have proved that there is no single universal diet, and the response to dietary inputs differs from person to person, which has a significant impact on the microbiome [16,17,59]. Although some host-derived factors are hard-wired and challenging to modulate, environmental factors such as dietary exposure can have a more significant impact on shaping the microbiome. It is now well-known that the microbiome can impact human physiology in various ways, including digestion, nutrient absorption, shaping the mucosal immune response, and synthesis [60].

Therefore, modulation of the microbiota through a personalized diet can be used to alter the host physiology and support medical and rehabilitation treatments towards maintenance or even regression of diseases. In this study, *Akkermansia*, *Prevotella*, and the Peptostreptococcaceae family seemed to be key factors modulated by diet in PKU subjects. According to the literature, dietary supplementation with the prebiotic inulin was observed to increase *Bifidobacterium* and *Akkermansia*, reducing neuroinflammation and anxiety and improving cognition in young and middle-aged mice [61,62]. The significance of diet in determining the alterations in the microbial community associated with the disease state cannot be underestimated. Omitting diet from the equation is crucial for comprehending the impact of IEM on the gut microbial community. Indeed, the PKU group exhibited underrepresentation in up to six pathways, which potentially impacts starch and sucrose metabolism and glycolysis/gluconeogenesis, as well as Phe, Tyr, Trp, valine, leucine and isoleucine biosynthesis, necessitating further investigation [32]. An in-depth analysis of specific pathways can facilitate identification of the reason for PKU patients’ heightened susceptibility to infectious diseases, including gastroenteritis, colitis, urticaria, and rhinitis [32]. 

This systematic review and meta-analysis has some limitations. Firstly, the selected studies exhibit substantial discrepancies in sociodemographic traits and dietary patterns among participants, thereby restricting their comparability and potentially impacting the consistency of the outcomes. In addition, there were variations in study quality, with the most significant quality concern being the absence of measures to control for confounding factors, posing a weakness to the robustness of the conclusions. However, this systematic review and meta-analysis is an initial effort to systematically characterize the different biodiversity in gut microbiota in people with PKU, and it opens new perspectives for future research to characterize the changes in bacterial composition that accompany different disease states and the corresponding expression patterns in both microbial and host genes.

## Figures and Tables

**Figure 1 ijms-24-17428-f001:**
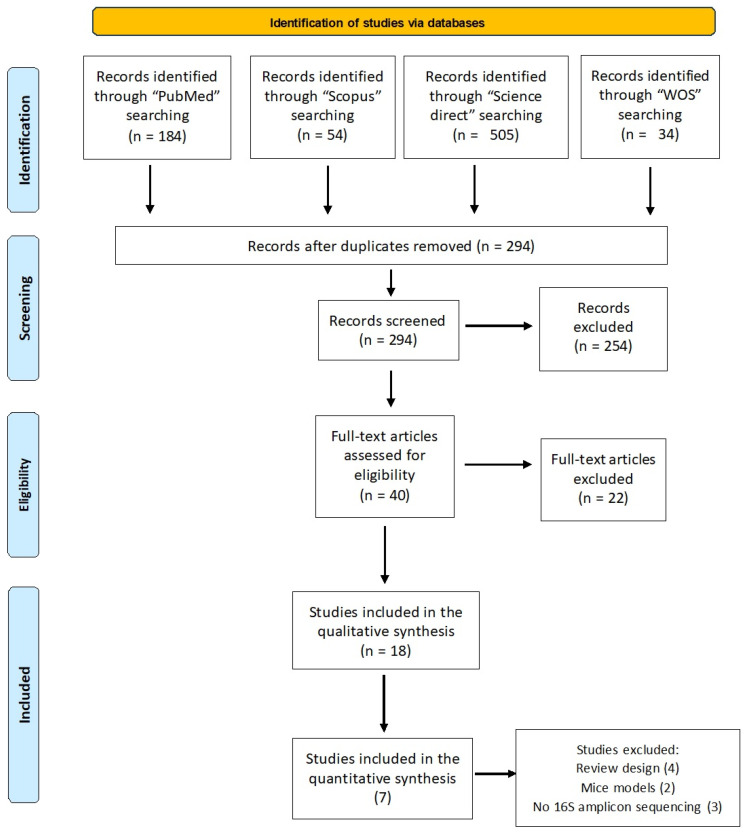
Flow chart of the search strategy, following the Preferred Reporting Items for Systematic Reviews and Meta-Analyses guidelines [21].

**Figure 2 ijms-24-17428-f002:**
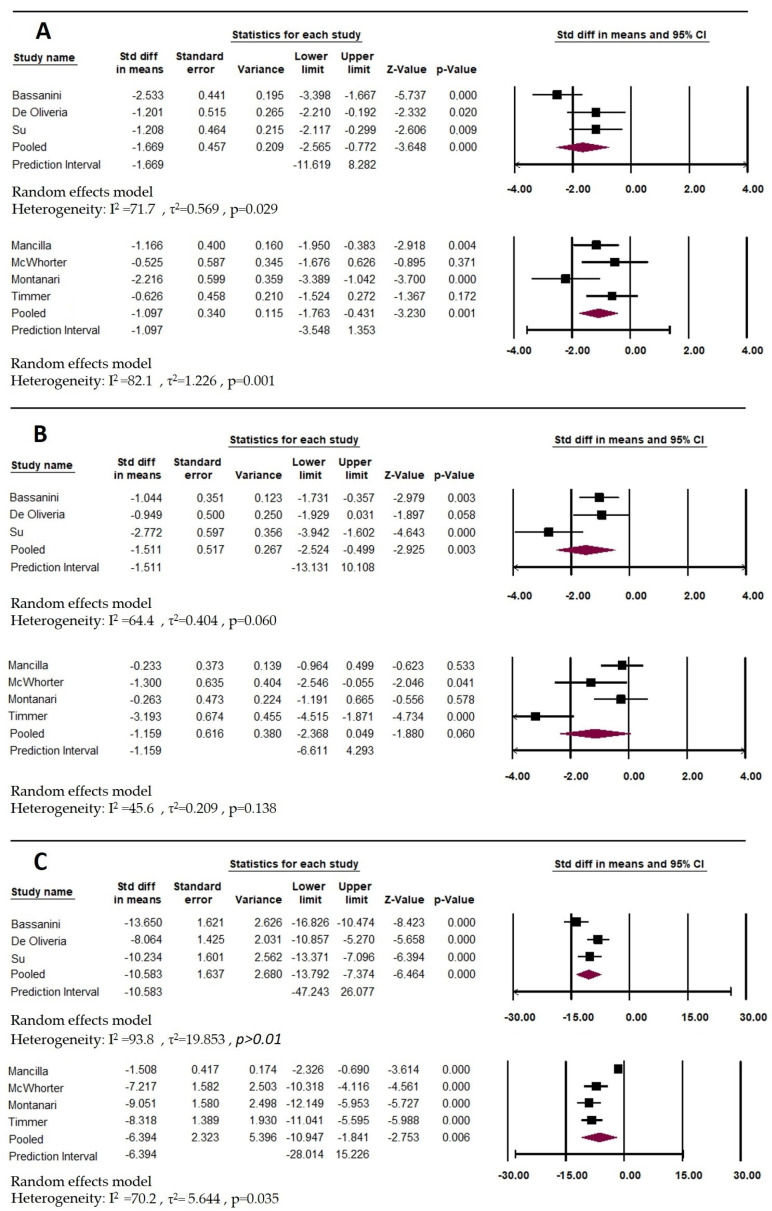
Forest plots for the differences in the means of the Bacillota/Bacteroidota ratio (formerly Firmicutes/Bacteroidetes ratio), Shannon, and OTU. (**A**) Shannon; (**B**) Bacillota/Bacteroidota ratio; (**C**) OTU.

**Table 1 ijms-24-17428-t001:** Summary of the original studies and main results related to PKU and the microbiota included in the qualitative study. * The quality scores assigned to each study are given in Appendix A.

Author, Year, Country, [Ref]	Type of Study	Aim of the Study	Organisms Target	Sample Characteristics	Blood Phe Levels	Treatment/Nutritional Control/Type of Diet	Study Quality *	Findings
Al-Zyoud et al., 2019, Jordan, [29]	Retrospective study	To compare the microbial profile obtained by traditional bacterial culture of the intestines of PKU patients and controls and to measure the effect of a low protein diet on the microbiota.	Human Childhood	Cases (PKU): 5 subjects Average age: 5 years Gender: 3 M, 2 F Controls: 20 subjects BMI z score: 3 ± 0.3; Average age: 4 years Gender: 11 M, 9 F BMI z score: 2.5 ± 0.3	No information	Dietary prebiotics: Cases: 5 yes, 0 no Controls: 13 yes, 7 no Probiotics: Cases: 0 yes, 5 no Controls: 8 yes, 12 no Breast milk feeding: Cases: 3 yes, 2 no Controls: 7 yes, 13 no Solid food: Cases: 2 yes and 3 no Controls: 8 yes and 12 no	Poor (3)	A statistically significant (*p* < 0.01) difference in *E. coli* presence in the gut flora of PKU subjects vs. control children was reported
Bassanini et al., 2019, Italy, [30]	Retrospective study	To study the impact of a Phe-restricted diet on the gut microbiota and possible consequences on the well-being of PKU patients	Human Childhood	PKU group: 21 subjects Average age: 10 ± 3.5 years Gender: 10 M, 11 F BMI z score: 0.2 ± 1; MHP group: 21 subjects Average age: 8 ± 3.4 years Gender: 12 M, 9 F BMI z score: 0.5 ± 1.1	Blood Phe levels: PKU group: 228 ± 87 μmol/L MHP group: 263 ± 97 μmol/L	Diet: PKU group: Carbs (%) 61.0 ± 7.0; Fibre (grams) 16.0 ± 9.1; Protein (%) 43.2 ± 15.1; Lipids (%) 29.6 ± 6.6; All of the subjects started diet/therapy at diagnosis of the disease, usually within 10 days of birth. MHP group: Carbs (%) 56.0 ± 5.9; Fibre (grams) 8.9 ± 2.6; Protein (%) 52.1 ± 11.1; Lipids (%) 32.6 ± 4.7	Good (7)	Although there was no significant difference in alpha diversity between the PKU and MHP groups, their gut microbiota analysis showed a marked separation. Notable variations were observed within the Firmicutes phylum. Despite the PKU children consuming more vegetables and fibre, similar to a vegan diet, their gut microbial profile differed from what is commonly observed in individuals following a high-fibre/low-protein diet. The depletion of useful microorganisms, such as *F. prausnitzii*, was noticed in the gut microbiota of PKU individuals. This suggests that the quality and quantity of carbohydrates consumed play a significant role in determining the observed shifts in Firmicutes within the PKU population.
MacDonald et al., 2011, UK, [31]	Exploratory pilot study	To study the influence of adding a specific proprietary blend of prebiotic oligosaccharides to a protein substitute suitable for infants with phenylketonuria	Human Childhood	Subjects: 9 infants with PKU Average age (y): 7.86 weeks (range 7.14–19.43) Gender: 3 M, 6 F BMI: not reported	No information	Infant Phenylalanine-free protein substitute containing prebiotic (scGOS/lcFOS [9:1 ratio])	Fair (5)	Treated infants showed phenylalanine levels in line with that observed with traditional therapy without prebiotics and higher levels of Bifidobacteria and lower stool pH.
Mancilla et al., 2021, USA, [32]	Descriptive study	To characterize the microbiota of PKU patients following a dietary treatment compared to controls	Human Adults	Cases (PKU): 11 subjects Average age: 33 ± 1.98 Gender: 6 M, 4 F (1 not reported by participant) Healthy controls: 21 subjects Average age: 29 ± 3.07 Gender: 11 M, 10 F	No information	No information	Fair (5)	Gut microbiota diversity was lower in PKU individuals (*p* < 0.001)
McWhorter et al., 2022, USA, [33]	Descriptive study	To compare differences in dietary treatment and gut microbiota in adult patients on a PKU liberalized diet and patients on pegvaliase-pqpz therapy	Human Adults	PKU group 1 (Traditional treatment): 6 subjects Average age: 28 (range: 23–40) Gender: 2M, 4F BMI: 30.4 (24.2–35.2) PKU group 2 (pegvaliase-pqpz treatment): 6 subjects Average age: 31 (range: 28–40) Gender: 3 M, 3 F BMI: 29.9 (22.1–39.6)	Mean blood Phe levels (μmol/L): Group 1: 314 (range: 157–468) Group 2: 50 (range: 0–314)	Diet Group 1: Calories (Kcal) 1943 ± 600; Carbs (g) 265.4 ± 103.6; Fibre (g) 16 ± 9.4; Total protein (from natural and medical foods) (g) 88.3 ± 19.6; Protein from foods 24.6 ± 5.0; Lipids (g) 59.9 ± 24.3 Increased intake of many micronutrients, including tyrosine, due to supplementation in medical foods. Group 2: Calories (Kcal) 2220 ± 556; Carbs (g) 290 ± 48.1; Fibre (g) 15.8 ± 6.8; Total protein 78.0 ± 24.9; Protein only from natural foods 78.0 ± 24.9; Lipids 85.6 ± 32.8	Good (7)	Individuals with PKU who receive treatment using pegvaliase -pqpz and follow a more flexible diet display noticeable variations in their dietary makeup compared to those who receive conventional Phe-restricted diets. Those in the traditional group exhibited a higher level of Verrucomicrobia phylum and Lachnobacterium genus, whilst the pegvaliase-pqpz group showed a greater presence of the *Prevotella* genus (*p* < 0.05).
Montanari, et al. 2022, Italy, [23]	Prospective study	To evaluate the effect of 6 months GMP supplementation on the intestinal microbiota of PKU patients by comparing their bacterial composition and clinical parameters before and after the intervention	Human Childhood and Adults	Cases (PKU): 9 patients (4 adults, 5 children) Classic PKU: 5 subjects Mild PKU: 4 subjects Average age: 20 years (range 7–38) Gender: 7 M, 2 F Average BMI (Kg/m^2^): 18.1 ± 1.5 Average BMI z score in paediatric populations: −0.2 ± 1.7	Mean Phe levels within the range [i.e., 120–360 μmol/L in childhood (<12 years) and 120–600 μmol/L in adolescence and adult age (>12 years)]	Restriction of natural proteins, use of amino acid supplements without Phe, low protein foods in order to reduce Phe intake according to their tolerance. No subject had ever used BH4 tetrahydrobiopterin	Good (8)	GMP seems to be safe from both the microbiological and the clinical point of view. A prebiotic effect of *Agathobacter* spp. and *Subdoligranulum*, the butyrate-producer, was observed. Phenylalanine values were kept below the age target and nutritional parameters.
Pinheiro De Oliveira et al., 2016, Brazil, [34]	Descriptive study	To characterize the microbiota of PKU patients following a dietary treatment compared to controls	Human Childhood	Cases (PKU): 8 subjects Average age: 4.24 ± 1.74 Gender: 6 M, 2 F BMI: 18.48 ± 1.30 Healthy controls: 10 subjects Average age: 6.06 ± 1.78 Gender: 4 M, 6 F BMI: 16.87 ± 1.55	Phe blood levels: Cases: PKU 307 μmol/L	Diet: Cases: Calories (Kcal) 1227.92 ± 187.91; Carbs (g) 215.08 ± 39.65; Fibre (g) 17.44 ± 3.42; Protein (g) 79.12 ± 15.41; Lipids (g) 16.12 ± 3.35 Phe-restricted diet supplemented with a prebiotic-free metabolic formula (protein substitute). Controls: Calories (Kcal) 1277.56 ± 78.43; Carbs (g)159.11 ± 9.44; Fibre (g) 12.37± 0.97; Protein (g) 72.82 ± 15.21; Lipids (g) 37.75 ± 4.45	Good (7)	Distinct taxonomic groups are present within the gut microbiome of PKU patients, which may be modulated by their plasma Phe concentration but it is not clear if these findings are a result of the disease itself or the modified diet.
Sawin et al., 2015, USA, [35]	Prospective study	To determine the prebiotic properties of GMP by characterizing caecal and faecal microbiota and caecal concentrations of SCFAs from PKU and wild-type (WT) mice fed diets containing GMP as the primary protein source	Mice	Experiment 1: Wild-type mice (WT): n = 45 Pahenu2 mice (PKU): n = 70 Divided into 3 groups: mice fed high-Phe casein, mice fed low-Phe AA, and mice fed low-Phe GMP Experiments 2 and 3: PKU mice fed AA diet (n = 6) PKU mice fed GMP diet (n = 8)	No information	High Phe level: casein administration (20% casein + 0.3% L-cysteine); low Phe level: supplementation with AA (17.5% free AA); GMP treated (20%)	Poor (4)	Functional foods may be beneficial in the management of PKU, as well as obesity and IBD.
Su et al., 2021, China, [36]	Retrospective study	To determine the characteristics of the microbiota in Ugur patients with PKU; to examine the correlation between clinical PKU phenotypes; to compare the microbial composition of PKU and healthy subjects	Human Childhood	Cases (PKU): 11 cases Mean age (y): 4.1 ± 0.7 (range: 0.8–8.25) Gender: 4 M, 7 F Average BMI (kg/m^2^): 16.36 ± 0.35 Controls: 11 sex-, nation-, and age-matched subjects	Mean blood Phe level (µmol/L): Cases: 481 ± 123 [pre-treatment: 986 ± 168]	Diet: Average daily carbohydrates: 192.36 ± 16.37; Average daily protein 33.46 ± 3.26 Cases: All patients followed a Phe-restricted diet supplemented with a metabolic formula (protein replacement) without prebiotics, and none received tetrahydrobiopterin (BH4), LNAA, or GMP treatment. Recommended daily intake of Phe was 135–330 mg according to guidelines for Chinese populations	Fair (5)	The PKU group showed very low levels of the *Bacteroides* genus, which was negatively associated with the blood Phe level. The authors propose that the reduced ability to degrade glycans in the PKU group may be due to the capacity of *Bacteroides* to break down complex and resistant glycans.
Timmer et al., 2021, Holland, [37]	Retrospective study	To evaluate and compare the faecal microbiota composition of patients with PKU and healthy controls; to study the influence of protein restriction and AA supplementation on the microbiome composition by comparing results between controls and PKU	Human Adults	PKU: 10 subjects Mean age (y): 35.5 (range: 19–50) Average BMI (kg/m^2^): 24.3 Controls: 10 subjects Mean age (y): 35.5 (range: 20–58) Average BMI (kg/m^2^): 23.9	No information	Diet: Group 1: Calories (Kcal) 1987, Phe intake (g) 31, Carbohydrates (g) 328, Lipids (g) 48, Protein from daily supplemental amino acids (g) 11 Group 2: Calories (Kcal) 1820, Phe intake (g) 19, Carbohydrates (g) 219, Lipids (g) 54, Protein from daily supplemental amino acids (g) 60 Group 3: Calories (Kcal) 1985, Phe intake (g) 83, Carbohydrates (g) 198, Lipids (g) 79.5, Protein from daily supplemental amino acids (g) 0	Fair (5)	The gut microbiome of PKU patients following a low-protein diet with added amino acids displayed lower diversity when compared to healthy adults not following a specific diet.
Van der Goot et al., 2022, Holland, [38]	Prospective study	To study the microbiota of mice with PKU fed diets with different protein contents	Mice	Wild type: n = 14; 7 M, 7 F Pah_enu2 (PKU model mice): n = 42 divided into 3 groups Pah_enu2 group 1—normal diet (high levels of Phe); n = 14; 7 M, 7 F Pah_enu2 group 2—Phe-liberalized diet (33% natural protein restriction); n = 14; 7 M, 7 F Pah_enu2 group 3—Phe-restriction (75% natural protein restriction); n = 14; 7 M, 7 F	No information	Wild-type and Pah_enu2 group 1 mice: baseline diet including 124 g/kg of dietary protein Pah_enu2 group 2: casein reduced by 33% and compensated by a blend of synthetic amino acids (Phe-free) * Pah_enu2 group 3: casein reduced by 75% and compensated by a blend of synthetic amino acids (Phe-free) * * Due to the reduced absorption of synthetic amino acids compared to protein, a putative protein conversion factor was taken into account, and an extra 20% amino acid blend was added	Good (7)	PKU leads to an altered gut microbiome composition in mice, which is least severe on a liberalized Phe-restricted diet
Verduci et al., 2018, Italy, [39]	Retrospective study	To compare dietary intake, gut microbiota diversity, and the production of short-chain fatty acids in children with PKU following a low-phenylalanine (Phe) diet and in children with mild hyperphenylalaninemia (MHP) following an unrestricted diet.	Human Adults	PKU group: 21 subjects Gender: 10 M, 11 F MHP group (control): 21 subjects Gender: 10 M, 11 F Average age (y): 8.69 (±3.57)	PKU cases: Mean Phe levels at baseline (µmol/L): 263 ± 97	PKU cases: Total protein (g) 43.2 (±15.1); Total formula Protein (g) 28.7 (±13.0); Carbs (g) 254.2 (±72.7); Fibre (g) 16.0 (±9.1); Lipids (g) 54.9 (±16.0) 13 different Phe-free, amino acid substitute medical foods (AA-MFs): most AA-MFs contained carbs (11 of 13), vitamins (10 of 13), minerals (10 of 13), and lipids (7 of 13); 3 of 13 supplemented with fibre. None contained probiotics. None of the subjects participating in the study were taking GMP and no patients were on tetrahydrobiopterin (BH4) or LNAA therapy	Poor (4)	The low-Phe diet, characterized by a higher carbohydrate intake, increases glycaemic load, resulting in a different quality of substrates for microbial fermentation. Further analyses, thoroughly evaluating microbial species altered by PKU diet are needed to better investigate gut microbiota in PKU children and supplements.

**Table 2 ijms-24-17428-t002:** Characteristics of the samples and specificities of the studies included in the meta-analysis.

Author, Year, Country, [Ref]	Sequencing Platform	Sequencing Region	Primer Used	Number of Average Reads Obtained per Sample	Passing Filter
Bassanini et al., 2019, Italy, [30]	MiSeq platform Illumina (2 × 251 base paired-end reads)	Region V3–V4 of the 16S rRNA gene	Illumina 16S rRNA custom primers	159,914	99%
Mancilla et al., 2021, USA, [32]	MiSeq platform Illumina (2 × 251 base paired-end reads)	Region V4 of the 16S rRNA gene	Primer pair 515F F-806R 16S rRNA	Not available	Not available
McWhorter et al., 2022, USA, [33]	Micro MiSeq platform Illumina (151 × 12 × 151 bp)	Region V4 of the 16S rRNA gene	Primers 515F/806R that directly amplify Bacteria/Archaea	179,009	99%
Montanari et al., 2022, Italy, [23]	MiSeq platform Illumina (2 × 251 base paired-end reads)	Region V3–V4 of the 16S rRNA gene	Illumina 16S rRNA custom primers	15,200	99%
Pinheiro de Oliveira et al., 2016, Brazil, [34]	Ion Torrent	Region V4 of the 16S rRNA gene	Primers 515F/806R that directly amplify Bacteria/Archaea	5890	99%
Su et al., 2021, China, [36]	MiSeq PE300 or NOVAseq PE250 platform Illumina	Region V3–V4 of the 16S rRNA genes	Primer pair 338F-806R 16S rRNA	16,186	99%
Timmer et al., 2021, Netherlands, [37]	MiSeq platform Illumina (2 × 251 base paired-end reads)	Region V3–V4 of the 16S rRNA genes	Illumina 16S rRNA custom primers	25,000	99%

**Table 3 ijms-24-17428-t003:** The overall results related to gut microbiota variability and structure in the seven studies included in the meta-analysis. Differences are shown in phyla/genera, and trends are represented in PKU subjects with respect to the control.

Author, Year, Country, [Ref]	Biodiversity Indicators Alpha Diversity	Biodiversity Indicators Beta Diversity	F/B	*Bacillota*	*Bacteroidota*	*Pseudomonadota*	*Actinomycetota*	*Verrucomicrobia*
Bassanini et al. 2019, Italy, [30]	↔ Shannon, OTUs, Chao1	Microbial communities are statistically different	↑	↑* ↑* *Clostridium* ↑* *Blautia* ↑* *Ruminococcus* ↑* *Lachnospiraceae (other)* ↓* *Faecalibacterium* ↓* *Veillonellaceae*	↓*	↑*	↑*	↓*
Mancilla et al., 2021 [32]	↓ Shannon, OTUs	Microbial communities are different	↓	↓ ↓ Lactobacillaceae ↓ *Blautia*	↑ ↑ *Bacteroides* ↑ *Alistipes*	↑	↓	↔
McWhorter et al., 2022, USA, [33]	↑* Shannon, Chao1, Simpson	Microbial communities are similar	↑*	*↑** *↑Clostridium* ↑* *Blautia* ↑* *Coprococcus* ↑* *Faecalibacterium*	↓* ↓* *Alistipes* ↓* *Prevotella*	↑*	↑	↑* ↑* *Akkermansia*
Montanari et al., 2022, Italy, [23]	↓ Shannon, Chao1 ↓ OTUs	Microbial communities are similar	↓*	↔ ↓* *Faecalibacterium*	↓* ↓* *Prevotella*	↑*	↑* ↑* *Bifidobacterium*	Not available
Pinheiro de Oliveira et al., 2016, Brazil, [34]	↓* Shannon, OTUs, Chao1	Microbial communities are statistically different	↓*	↓ * ↓* *Clostridium* ↓* *Coprococcus* ↓* *Ruminococcus* ↓* *Lachnospiraceae* ↓* *Veillonella* *↓ Erysipelotricaceae*	↑* ↑* *Prevotella*	↓*	ND	↑* ↑* *Akkermansia*
Su et al., 2021, China, [36]	↓* Shannon, Chao1	Microbial communities are statistically different	↑*	↑ ↓* *Faecalibacterium*	↓* ↔ Prevotella	↑*	↑* ↑* *Bifidobacterium* ↑* *Collinsella*	Not available
Timmer et al., 2021, Netherlands, [37]	↓* Shannon, OTUs	Microbial communities are statistically different	↑*	↑* ↑* *Clostridium* ↑* *Ruminococcus* ↓* *Veillonellaceae*	↓*	↑*	↑*	↓*

↑ = increase; ↓ = decrease; ↑* = statistically significant increase; ↓* = statistically significant decrease; ↔ = no differences between groups.

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
