# Peer review of "Systematic Review and Meta-Analysis of Dietary Interventions and Microbiome in Phenylketonuria"

_ijms, 2023, doi:10.3390/ijms242417428_

Round 1

Reviewer 1 Report

Comments and Suggestions for Authors

The first phrase of the title does not need to be included unless this is anticipated to be one of a series of articles examining the microbiome of various inborn errors of metabolism.  Writing out phenylketonuria instead of leaving it as PKU in a shortened title is also needed.

 Abstract comment ‘highlights a role for the type of diet’ needs to be explained.

Introduction: Phenylketonuria is not only caused by mutations in the PAH gene but it is also caused by cofactor synthetic defects; your paragraph beginning on line 46 needs to reflect this.

To be inclusive about the therapies available for PKU, you need to include pegvaliase pqpz therapy in the paragraph beginning on line 67.  This becomes essential later on in the manuscript since some of your included studies compare patients treated with this phenylalanine ammonia lyase enzyme preparation.

Any source of intact protein can and is used in the management of hyperphenylalaninemias.  It is not just milk protein.

Large neutral amino acid therapy (line 91 close (has a role in interfering with the absorption of phenylalanine across the intestinal tract and in altering the amount of phenylalanine that gets into the CNS cross the blood-brain barrier.  The mechanism of action is very different than that of glycomacropeptide.

The paragraph beginning on line 93 needs to be rewritten or placed in a different part of the introduction.  It fits best with some of the concepts in the first paragraph of the introduction.  You have gone from a several paragraph discussion of PKU back to a more generalized statement.

In the last paragraph of the introduction, you do not explain to the reader how you plan to increase the knowledge of the structure of the microbiota nor do you define a personalized dietary intervention or the targets for such an intervention.  This severely undermines expectations for the rest of the manuscript.

Section 2: On line 115 you need to define energy drinks.  Liquid medical food products used for the treatment of inborn errors of metabolism would not be considered "energy drinks".

You need to clarify the language used on line 140 regarding inception.  Do you mean from 2011, the publication date of one of the first papers used, the beginning of any of the databases that were used for the search or the beginning of the science on phenylketonuria.  That science extends back as far as 1934 and publications about dietary therapy since the late 1950s

Line 144 is confusing this paper is focused on phenylketonuria.  What other type of congenital error of metabolism, more commonly known as an inborn error of metabolism, were you extracting?

Be careful with your statistics as the I2 test has a substantial bias when the number of studies is small.

Clarify line 175 as there is not a high volume of samples but rather a very low number of final included publications.

The first paragraph of section 3 Results and Discussion is essentially a recapitulation of the introduction.  It needs to be rewritten in order to be a more forceful opening.  As an example: While the purpose of dietary treatment of inborn errors of metabolism is to reduce the target toxic compound(s), such therapy may also cause dangerous deficiencies of nutrient groups.  Furthermore, what you mean by "to draw therapeutic dietary interventions".  The concerns of this reviewer in regard to the remainder of the paragraph, have been expressed above.

Since this paper focuses on phenylketonuria, why were references 3, 11, 16, 12 included in table 1?  If none of the reviews has a novel data relevant for PKU they should have been further excluded very briefly in the methods section without the inclusion of table 2.  At best, table 2 should be in supplemental material.

Table 1 should separate objective findings from published opinion as separate columns.  This may require reorientation of the table format to be able to include this number of columns.  Table should also be revised so that the studies on animals are grouped followed by another criterion such as age of subjects so that the reader is better able to compare in contrast with populations that were the subject of your data.  Since Macdonald et all studied infants, they have no ability to comply with the protocol requirements, it is their parents who would have been complying.  Precision is key in a meta-analysis.  Ages should be designated in years where relevant such as in the inclusion of reference 29.  Why is the type of childbirth relevant for dietary intervention, especially multiple years after birth?  Exclude any extraneous information from the table.  Your entry for reference 29 has a serious error or in that blood phenylalanine concentrations were 307.33 µmol/L which in a small group and given the potential precision of the assays should have been rounded to 307.  One of the patients in that group was being breast-fed which might change the results of which species were detected in the stool sample.  In reference 26 were the individuals really consuming 2542 g of carbohydrate which would represent over 6000 kcals per day?  You have typographical errors in the following column as ‘glycemic’ should be used.  There is a recurrent typographical error in the entries for references 25 and 28 in the spelling of ‘retrospective’. In Bassanini et al., there is a shortcoming in the listed data as percent protein is not the same as grams of protein being given nor is it the same as the missing data from most of these which is the number of milligrams or micromoles of phenylalanine provided per kilogram of body weight.  It is the milligrams per kilo which is most commonly used within the clinical setting to measure tolerance of intact protein.  The entry for Timmer et al could be shortened as the information about the UCD patients is not relevant for the rest of your manuscript.  Use the accepted generic name pegvaliase pqpz as Palynziq is a registered trademark and must be denoted as such in every use. Since pegvaliase pqpz is intended to allow better control and often higher protein tolerance, the differences in diet composition reported in McWhorter et al are expected. In Montanari et al., the information that phenylalanine concentrations were kept below the age-target does not have relevance as a finding but has relevance under the type of diet.  Since different providers use different targets, more detailed information about what the actual levels were, relevant to the number of milligrams of phenylalanine consumed, is needed.  However, the information about GMP being safe is not relevant to your meta-analysis.  There is also a typographical error in the sample characteristics ‘patients’.

The mouse model Pah enu2 is generally denoted with a supercripted enu2 or with a space.  The findings that PKU leads to an altered get microbiome compensation needs to be critically questioned.  Is it the PKU or is it the phenylalanine restriction or altered composition of the blend of synthetic/semisynthetic amino acid food product that is responsible for the alterations?

Lines 218-220 appear to be a duplicate of lines 204-206.

Is there a reason that the studies that composed table 3 are presented in different order in table 4?

You need to produce some evidence that third and fourth decade aged adults have a different microbiota before the statement on lines 254-255.  Otherwise you need to make a separate paragraph beginning with the information on line 253 as a disclaimer for the quality of information for the studies in the meta-analysis.  How is reference 39 relevant for PKU with a span of approximately 15 years across the adult subjects in the cited paper(s)?  Furthermore how can you do a meta-analysis across the age groups if the microbiome of children is different than adults (your reference 47)?  You would then have to do a separate analysis of those studies involving individuals under 18 years of age, potentially those involving infants and those of involving adults which would lead to too small a number of studies to draw any conclusions.

Figure 2 in a high-quality PDF is not readable for the data tables.

Is 43±15 g protein intake different from 52±11 g of protein given the size of the sample groups in reference 28?  If it is not, then the statement on lines 286-287 is not supported by the evidence that you present.

In lines 299-305 you need to account for the information in the table that the BMI Z score in the individuals with PKU was 3.1, suggesting significant obesity compared to their healthy controls.

The statements in line 313-316 may be better placed adjacent to your comments in lines 300-301.

In Su et al, there is a significant range of blood phenylalanine concentrations which would suggest that the majority of individuals were not in good dietary control during this study.  Most of Western Europe would use blood phenylalanine concentrations less than 360 µmol/L as evidence of good metabolic control for this age group.  Values above 1200 µmol/L would indicate no compliance with an intervention for the classical PAH deficiency form of phenylketonuria.  The quality of this data needs to be considered before doing any meta-analysis.

Are lines 352-354 a conclusion or does this need a reference as this statement is similar to the point of Kirby et al. 2019?

Since Mancilla et al is included as a reference, why was this not included as one of the studies in your meta-analysis?

You need to point out in your discussion section where the deficiencies in the studies impede analysis and meta-analysis as well as what can/should be done to improve the quality of the data so that we will have the most scientifically sound conclusions about the effect of PKU and intervention for that disease have on our intestinal symbionts.

Comments on the Quality of English Language

A number of typographical errors and grammar questions were noted as more minor concerns with this manuscript.  Among them are: 

Line 41: phenylketonuria should be lowercase.

Line 64: Typographical error: severe.

Line 126: Capitalize September.

Line 131: Grammatical change: You have not said the criteria, you have stated the criteria.

Line 134: A comma is needed between reviews and randomized.

Line 140: September needs to be capitalized.

The grammar needs to be changed in the lines 145-146 as the reader and the authors are interacting in real-time; this should be "limitations of the study are reported".

The sentence beginning on line 170 needs to be rewritten for clarity in English.

Table 1: McWhote et al. typographical error in ‘group’

Table 3: typographical error ‘paired’ not paried with 3 occurrences.

Line 228: Typographical error: ‘Archaea’

On line 238: OUTs is introduced without an explanation that this is operational taxonomic unit.  It is suggested to write this out for the first occurrence.

Line 250: Include the et al. after Bassanini.

Line 261: Correct the typographical error of the first author of reference 30.  The same error is perpetuated in line 273.  Like the comment above, the et a.l needs to be included after both studies’ first author in line 261

Other items of typography or grammar are in the section above 

Author Response

Review 1

Comments and Suggestions for Authors

The first phrase of the title does not need to be included unless this is anticipated to be one of a series of articles examining the microbiome of various inborn errors of metabolism.  Writing out phenylketonuria instead of leaving it as PKU in a shortened title is also needed.

R: We appreciate the reviewer's detailed and valuable suggestions. After carefully reviewing all the suggestions, we believe our work has significantly improved. We have changed the title as suggested.

Abstract comment ‘highlights a role for the type of diet’ needs to be explained.

Thank you for providing us with the opportunity to clarify our meaning. We have revised the abstract section.

Introduction: Phenylketonuria is not only caused by mutations in the PAH gene but it is also caused by cofactor synthetic defects; your paragraph beginning on line 46 needs to reflect this.

R: Thanks for the observations. We have modified the sentence to make it clearer and more complete. The paragraph has been modified.

To be inclusive about the therapies available for PKU, you need to include pegvaliase pqpz therapy in the paragraph beginning on line 67.  This becomes essential later on in the manuscript since some of your included studies compare patients treated with this phenylalanine ammonia lyase enzyme preparation.

R: Thanks. We have added and completed the list of treatments. The text has been modified as follows: “Some patients with BH4 responsiveness can be treated with sapropterin dihydrochloride (Kuvan, BioMarin Corporation, Tiburon, CA), which is a pharmaceutical chaperone. Moreover, FDA approved a new therapy in 2018 for adults in the USA and for patients aged 16 years or older in Europe. This therapy is based on a bacterial enzyme that can break down Phe (PEGVALIASE PQPZ, BioMarin Pharmaceutical Inc., USA) [4]. Additionally, enzyme replacement and gene therapy are potential future treatments [7, 8]”

Any source of intact protein can and is used in the management of hyperphenylalaninemias.  It is not just milk protein.

R: Thanks for the comment. We have modified the sentence to make it clearer and more complete. We also added the Phenylalanine range. The text has been modified as follows: “Phenylalanine is essential for protein synthesis and therefore a minimal amount must be provided through the diet to support tissue growth and repair during childhood and tissue repair in adulthood, maintaining plasma phenylalanine concentrations within recommended ranges (2–6 mg/dL or 120–360 μmol/L) [8-11]."

Large neutral amino acid therapy (line 91 close (has a role in interfering with the absorption of phenylalanine across the intestinal tract and in altering the amount of phenylalanine that gets into the CNS cross the blood-brain barrier.  The mechanism of action is very different than that of glycomacropeptide.

R: Thanks for the suggestion. We have revised the paragraph following the suggestions of the reviewer. 

The paragraph beginning on line 93 needs to be rewritten or placed in a different part of the introduction.  It fits best with some of the concepts in the first paragraph of the introduction.  You have gone from a several paragraph discussions of PKU back to a more generalized statement.

R: We have revised all paragraphs in accordance with the suggestions of the reviewer and focusing on PKU.

In the last paragraph of the introduction, you do not explain to the reader how you plan to increase the knowledge of the structure of the microbiota nor do you define a personalized dietary intervention or the targets for such an intervention.  This severely undermines expectations for the rest of the manuscript.

R: Thank you for offering the opportunity of better clarifying what we mean, we have defined and specification the aim and specific purpose.

Section 2: On line 115 you need to define energy drinks.  Liquid medical food products used for the treatment of inborn errors of metabolism would not be considered "energy drinks".

R: Thanks for the comment. This sentence was modified because there was a mistake, we have modified the point and defined the outcomes: “Outcome: the abundance of phyla, as the Bacillota/Bacteroidota ratio, and alpha diversity and beta-diversity indices”.

You need to clarify the language used on line 140 regarding inception.  Do you mean from 2011, the publication date of one of the first papers used, the beginning of any of the databases that were used for the search or the beginning of the science on phenylketonuria.  That science extends back as far as 1934 and publications about dietary therapy since the late 1950s

R: Thanks, we have modified the sentence to improve the clarity of text. We have interrogated four electronic databases (PubMed, Scopus, Science direct and Web of Science), using the specific terms [(“metabolism, inborn errors”[MeSH Terms] OR (“metabolism”[All Fields] AND “inborn”[All Fields] AND “errors”[All Fields]) OR “inborn errors metabolism”[All Fields] OR (“inborn”[All Fields] AND “errors”[All Fields] AND “metabolism”[All Fields])) AND ((“phenylketonurias”[MeSH Terms] OR “phenylketonurias”[All Fields] OR “phenylketonuria”[All Fields]) AND (“microbiota”[MeSH Terms] OR “microbiota”[All Fields] OR “microbiotas”[All Fields] OR “microbiota s”[ All Fields] OR “microbiotae”[All Fields] OR “microbiome”)]. The search was conducted from database inception until 30th September 2022, to ensure we do not miss any studies in the literature. However, available studies on gut microbiota and PKU date back only to the early 2000s.

Line 144 is confusing this paper is focused on phenylketonuria.  What other type of congenital error of metabolism, more commonly known as an inborn error of metabolism, were you extracting?

R: Thank you for giving us the chance to clarify our point. We made changes to the sentence to correct a mistake and to define the eligible record more precisely. We also removed the "type of congenital error of metabolism" in line with your suggestion and considering the subsequent changes made in the manuscript.

Be careful with your statistics as the I2 test has a substantial bias when the number of studies is small.

R: We thank the Reviewer for posing this important question, which offer us the opportunity for clarifying our point of view, although, of course, we absolutely respect the point of view advanced by the Reviewer. We agree with the Reviewer that I2 as a substantial bias when the number of studies is small and does not eliminate the uncertainty that comes from having a small number of studies. In general, no statistics can make this. In small meta-analyses, for the same reason that Q has low power, I2 is very imprecise. However, several meta-analyses with very few studies are very common [such as Żegleń, M., Kryst, Ł., & Bąbel, P. (2023). Diet, gym, supplements, or maybe it is all in your mind? A systematic review and meta-analysis of studies on placebo and nocebo effects in weight loss in adults. Obesity reviews: an official journal of the International Association for the Study of Obesity, 10.1111/obr.13660. Advance online publication. https://doi.org/10.1111/obr.13660; Friede,  T., Röver, C., Wandel, S., & Neuenschwander, B. (2017). Meta-analysis of few small studies in orphan diseases. Research synthesis methods, 8(1), 79–91. https://doi.org/10.1002/jrsm.1217; Wehrli, S., Rohrbach, M., & Landolt, M. A. (2023). Quality of life of pediatric and adult individuals with osteogenesis imperfecta: a meta-analysis. Orphanet journal of rare diseases, 18(1), 123. https://doi.org/10.1186/s13023-023-02728-z].  Moreover, there are guidelines to perform metanalysis in this case [Bender, R., Friede, T., Koch, A., Kuss, O., Schlattmann, P., Schwarzer, G., & Skipka, G. (2018). Methods for evidence synthesis in the case of very few studies. Research synthesis methods, 9(3), 382–392. https://doi.org/10.1002/jrsm.1297]. The metanalysis is possible using the guidelines published by a working group of the Cochrane Collaboration that recommended the use of the Knapp‐Hartung method for meta‐analyses with random effects. However, as heterogeneity cannot be reliably estimated if only very few studies are available, the Knapp‐Hartung method, while correctly accounting for the corresponding uncertainty, has very low power. In our case the goal of the study is to explore the structure of the microbiota in patients affected by PKU, analyzing the data coming from the available literature on this topic and estimating the modifications in the biodiversity indices among the gut microbiota in PKU patients respect to the control population, also focusing, when it was possible, on response to different treatments. However, the studies available on these aspects are truly small and recently published. But the secondary objective of our study is also to outline the fundamental points and Milestones to outline new study strategies in this issue.

Clarify line 175 as there is not a high volume of samples but rather a very low number of final included publications.

R: Thanks, we have modified the sentence to improve the clarity of text.

The first paragraph of section 3 Results and Discussion is essentially a recapitulation of the introduction.  It needs to be rewritten in order to be a more forceful opening.  As an example: “While the purpose of dietary treatment of inborn errors of metabolism is to reduce the target toxic compound(s), such therapy may also cause dangerous deficiencies of nutrient groups.  Furthermore, what you mean by "to draw therapeutic dietary interventions".  The concerns of this reviewer in regard to the remainder of the paragraph, have been expressed above.

R: Thanks, we have modified the sentence to improve the clarity of text.

Since this paper focuses on phenylketonuria, why were references 3, 11, 16, 12 included in table 1?  If none of the reviews has a novel data relevant for PKU they should have been further excluded very briefly in the methods section without the inclusion of table 2.  

At best, table 2 should be in supplemental material.

R: Thanks, we have modified the table. The Table 1 now included only the original articles included in qualitative analysis, and Tabel S2 contain the review articles.  We have improved Table 1 by removing papers that were not part of the qualitative analysis to improve clarity. Furthermore, we have included all reviews in Table S2, which now contains all reviews. Additionally, we have provided a clear breakdown of the quality score analysis, which can be found in Table S1. We have also revised Figure 1, enhanced its clarity and provided information on which articles were excluded, as well as the reasoning behind these decisions. This update was made in line with PRISMA guidelines [Haddaway, N. R., Page, M. J., Pritchard, C. C., & McGuinness, L. A. (2022). PRISMA2020: An R package and Shiny app for producing PRISMA 2020-compliant flow diagrams, with interactivity for optimized digital transparency and Open Synthesis Campbell Systematic Reviews, 18, e1230. https://doi.org/10.1002/cl2.1230].

Table 1 should separate objective findings from published opinion as separate columns.  This may require reorientation of the table format to be able to include this number of columns.  Table should also be revised so that the studies on animals are grouped followed by another criterion such as age of subjects so that the reader is better able to compare in contrast with populations that were the subject of your data.  Since Macdonald et all studied infants, they have no ability to comply with the protocol requirements, it is their parents who would have been complying.  Precision is key in a meta-analysis.  Ages should be designated in years where relevant such as in the inclusion of reference 29.  Why is the type of childbirth relevant for dietary intervention, especially multiple years after birth?  Exclude any extraneous information from the table.  

R: Thanks, table 1 was modified according to the reviewer's suggestion. Table 1 is now designed with only original papers included in qualitative analysis (total n. 12) while the reviewer is present in Table s2. We have excluded any extraneous information from the table.  

Below are the answers to each highlighted point.

Your entry for reference 29 has a serious error or in that blood phenylalanine concentrations were 307.33 µmol/L which in a small group and given the potential precision of the assays should have been rounded to 307.  

R: Thanks, we have modified the text.

One of the patients in that group was being breast-fed which might change the results of which species were detected in the stool sample.  In reference 26 were the individuals really consuming 2542 g of carbohydrate which would represent over 6000 kcals per day?  You have typographical errors in the following column as ‘glycemic’ should be used.  

R: Thanks, we have modified the information in the table and corrected the typographical error (both g of carbohydrate and glycemic). Considering the significant gaps in the subject selection process, we have assessed this work to have a low level of quality.

There is a recurrent typographical error in the entries for references 25 and 28 in the spelling of ‘retrospective’.

R: Thanks, we have modified the text.

 In Bassanini et al., there is a shortcoming in the listed data as percent protein is not the same as grams of protein being given nor is it the same as the missing data from most of these which is the number of milligrams or micromoles of phenylalanine provided per kilogram of body weight.  It is the milligrams per kilo which is most commonly used within the clinical setting to measure tolerance of intact protein.  

R: Thanks for your observations. We have modified the text to improve the clarity of the manuscript.

The entry for Timmer et al could be shortened as the information about the UCD patients is not relevant for the rest of your manuscript.  

R: Thanks, we have modified the table 1.

Use the accepted generic name pegvaliase pqpz as Palynziq is a registered trademark and must be denoted as such in every use. Since pegvaliase pqpz is intended to allow better control and often higher protein tolerance, the differences in diet composition reported in McWhorter et al are expected.

R: Thanks, we have corrected in Table and in the text.

In Montanari et al., the information that phenylalanine concentrations were kept below the age-target does not have relevance as a finding but has relevance under the type of diet.  Since different providers use different targets, more detailed information about what the actual levels were, relevant to the number of milligrams of phenylalanine consumed, is needed.  

R: Thanks, we have transferred the information into separate columns to enhance data clarity and amended errors in the table and text.

However, the information about GMP being safe is not relevant to your meta-analysis.  There is also a typographical error in the sample characteristics ‘patients’.

R: Thanks, we have corrected in Table and in the text.

The mouse model Pah enu2 is generally denoted with a supercripted enu2 or with a space.  The findings that PKU leads to an altered get microbiome compensation needs to be critically questioned.  Is it the PKU or is it the phenylalanine restriction or altered composition of the blend of synthetic/semisynthetic amino acid food product that is responsible for the alterations?

R: Thanks, we have corrected in Table and in the text.

Lines 218-220 appear to be a duplicate of lines 204-206.

R: Thanks, we have corrected checked and modified the text.

Is there a reason that the studies that composed table 3 are presented in different order in table 4?

R: Thanks, we have corrected in Table and listed the papers in alphabetical order

You need to produce some evidence that third and fourth decade aged adults have a different microbiota before the statement on lines 254-255.  Otherwise you need to make a separate paragraph beginning with the information on line 253 as a disclaimer for the quality of information for the studies in the meta-analysis.  How is reference 39 relevant for PKU with a span of approximately 15 years across the adult subjects in the cited paper(s)?  Furthermore how can you do a meta-analysis across the age groups if the microbiome of children is different than adults (your reference 47)?  You would then have to do a separate analysis of those studies involving individuals under 18 years of age, potentially those involving infants and those of involving adults which would lead to too small a number of studies to draw any conclusions.

R: Thanks for your suggestion and We analyzed only the data for adult subjects in our metagenomics analyses of this work.  Upon updating the work, Figures 2 revealed a different breakdown.

Figure 2 in a high-quality PDF is not readable for the data tables.

R: OK done, the figure was updated and improve the quality.

Is 43±15 g protein intake different from 52±11 g of protein given the size of the sample groups in reference 28?  If it is not, then the statement on lines 286-287 is not supported by the evidence that you present.

R: Thanks, we have modified the text. In according also the authors, PKU subjects results in a reduction in Bacteroides and Prevotella, the two main genera belonging to the Bacteroidetes This is already confirmed in the literature in several settings (Moreno-Pérez, D., Bressa, C., Bailén, M., Hamed-Bousdar, S., Naclerio, F., Carmona, M., Pérez, M., González-Soltero, R., Montalvo-Lominchar, M. G., Carabaña, C., & Larrosa, M. (2018). Effect of a Protein Supplement on the Gut Microbiota of Endurance Athletes: A Randomized, Controlled, Double-Blind Pilot Study. Nutrients, 10(3), 337. https://doi.org/10.3390/nu10030337; Wang, Y., Li, Y., Bo, L., Zhou, E., Chen, Y., Naranmandakh, S., Xie, W., Ru, Q., Chen, L., Zhu, Z., Ding, C., & Wu, Y. (2023). Progress of linking gut microbiota and musculoskeletal health: casualty, mechanisms, and translational values. Gut microbes, 15(2), 2263207. https://doi.org/10.1080/19490976.2023.2263207). We understood your point regarding the insufficient strength of the data regarding the difference in dietary amounts between the two groups. However, in agreement with the authors, it can be assumed that the changes in microbiota composition are due to the lower protein intake. In this regard, literature remains in agreement with this observation.

In lines 299-305 you need to account for the information in the table that the BMI Z score in the individuals with PKU was 3.1, suggesting significant obesity compared to their healthy controls.

R: Thank you, there are several mistakes in the table, we have corrected them.

The statements in line 313-316 may be better placed adjacent to your comments in lines 300-301.

R: Thank you for your suggestion, we have improved the text.

In Su et al, there is a significant range of blood phenylalanine concentrations which would suggest that the majority of individuals were not in good dietary control during this study.  Most of Western Europe would use blood phenylalanine concentrations less than 360 µmol/L as evidence of good metabolic control for this age group.  Values above 1200 µmol/L would indicate no compliance with an intervention for the classical PAH deficiency form of phenylketonuria.  The quality of this data needs to be considered before doing any meta-analysis.

A: Okay, thank you. We checked this article and updated the quality score. This suggestion was helpful in our discussion and added to the limit of our meta-analysis. Obviously, this is a very new topic and few papers have been performed. However, these works may have limitations in the design/selection of the population and also in the results. However, making a point about these aspects is essential. The purpose of our paper is to provide some insights into the possible key points in microbiota variation in individuals with PKU and to outline some factors that may modulate this variation.  The small number of papers and the heterogeneity of the studies is a strong limitation, we understand, but this very fact must give strength to the design of new work on this topic, and our contribution is only to outline some key points to be explored further.

Are lines 352-354 a conclusion or does this need a reference as this statement is similar to the point of Kirby et al. 2019?

R: Thanks. We have checked this paper and modified the discussion to avoid overlapping.

Since Mancilla et al is included as a reference, why was this not included as one of the studies in your meta-analysis?

R: Thanks. We initially did not include this paper because the authors did not include information related to the diet and the treatment. However, since we have modified our analysis and the discussion, we decided to included also this paper in the revised manuscript.

You need to point out in your discussion section where the deficiencies in the studies impede analysis and meta-analysis as well as what can/should be done to improve the quality of the data so that we will have the most scientifically sound conclusions about the effect of PKU and intervention for that disease have on our intestinal symbionts.

R: Thanks, the whole text has been reviewed in order to improve the quality and lead the clarification. Also, the discussion was improved according to the suggestions.

Comments on the Quality of English Language

A number of typographical errors and grammar questions were noted as more minor concerns with this manuscript.  Among them are: 

Line 41: phenylketonuria should be lowercase.

R: The text has been corrected accordingly

Line 64: Typographical error: severe.

R: The text has been corrected accordingly

Line 126: Capitalize September.

R: The text has been corrected accordingly

Line 131: Grammatical change: You have not said the criteria, you have stated the criteria.

R: The text has been corrected accordingly

Line 134: A comma is needed between reviews and randomized.

R: The text has been corrected accordingly

Line 140: September needs to be capitalized.

R: The text has been corrected accordingly

The grammar needs to be changed in the lines 145-146 as the reader and the authors are interacting in real-time; this should be "limitations of the study are reported".

R: The text has been corrected accordingly

The sentence beginning on line 170 needs to be rewritten for clarity in English.

Table 1: McWhote et al. typographical error in ‘group’

R: The text has been corrected accordingly

Table 3: typographical error ‘paired’ not paried with 3 occurrences.

R: The text has been corrected accordingly

Line 228: Typographical error: ‘Archaea’

R: The text has been corrected accordingly

On line 238: OUTs is introduced without an explanation that this is operational taxonomic unit.  It is suggested to write this out for the first occurrence.

R: The text has been corrected accordingly

Line 250: Include the et al. after Bassanini.

R: The text has been corrected accordingly

Line 261: Correct the typographical error of the first author of reference 30.  The same error is perpetuated in line 273.  Like the comment above, the et a.l needs to be included after both studies’ first author in line 261

Other items of typography or grammar are in the section above 

R: The text has been corrected accordingly

Reviewer 2 Report

Comments and Suggestions for Authors

The review by Francesca Ubaldi et al.,  aims to investigate the effect of PKU therapies on the intestinal microbiota by using meta-analysis. However, because these review based on the limited number of studies, it seems non-systematic. The meta-analysis seems insufficient. The discussion section is not in-depth, only describing a few literature without synthesizing them for discussion.

Title: change "PKU" to "phenylketonuria".

The content is repeated in Lines 218-220 and Lines 204-206.

Line 203: change "microbiota" to "gut microbiota".

The Figure 2 is unclear, it need a high definition.

Author Response

Review 2

Comments and Suggestions for Authors

The review by Francesca Ubaldi et al.,  aims to investigate the effect of PKU therapies on the intestinal microbiota by using meta-analysis. However, because these review based on the limited number of studies, it seems non-systematic.

R: We appreciate the reviewer's detailed and valuable suggestions. After carefully reviewing all the suggestions, we believe our work has significantly improved. We have changed the title as suggested.

The meta-analysis seems insufficient. The discussion section is not in-depth, only describing a few literature without synthesizing them for discussion.

R: We have corrected and improved the discussion.

Title: change "PKU" to "phenylketonuria".

R: We have corrected and improved the discussion.

The content is repeated in Lines 218-220 and Lines 204-206.

R: Thanks. The text has been corrected accordingly.

Line 203: change "microbiota" to "gut microbiota".

R: The text has been corrected accordingly

The Figure 2 is unclear, it need a high definition.

R: The text has been corrected accordingly

Round 2

Reviewer 2 Report

Comments and Suggestions for Authors

This is my second time to review this manuscript. In this manuscript, the author revised this manuscript in appropriate modifications based on my previous comments. Before it accepted, there are still have some minor comments need be revised as follow.

Line 49. What the mean of DNAJC12?

Line 251. change "CHAO1" to "Chao1".

Figure 1. The font size of the word in the figure is small. Please modificated.

Line 286. Change "Bacillota" to "Firmicutes".

Line 317. Blautia spp need italic.